# Comparison of Plaque Size, Thermal Stability, and Replication Rate among SARS-CoV-2 Variants of Concern

**DOI:** 10.3390/v14010055

**Published:** 2021-12-30

**Authors:** Gi Uk Jeong, Gun Young Yoon, Hyun Woo Moon, Wooseong Lee, Insu Hwang, Haesoo Kim, Kyun-Do Kim, Chonsaeng Kim, Dae-Gyun Ahn, Bum-Tae Kim, Seong-Jun Kim, Young-Chan Kwon

**Affiliations:** Center for Convergent Research of Emerging Virus Infections, Korean Research Institute of Chemical Technology, Daejeon 34114, Korea; jay7@krict.re.kr (G.U.J.); emdoc96@krict.re.kr (G.Y.Y.);moonhw@krict.re.kr (H.W.M.); ivemania@krict.re.kr (W.L.); hwang@krict.re.kr (I.H.); haiskim@krict.re.kr (H.K.); kdkim@krict.re.kr (K.-D.K.); chonskim@krict.re.kr (C.K.); dgahn@krict.re.kr (D.-G.A.); btkim@krict.re.kr (B.-T.K.)

**Keywords:** SARS-CoV-2, variants of concern, thermal stability, replication rate

## Abstract

SARS-CoV-2, like other RNA viruses, has a propensity for genetic evolution owing to the low fidelity of its viral polymerase. Several recent reports have described a series of novel SARS-CoV-2 variants. Some of these have been identified as variants of concern (VOCs), including alpha (B.1.1.7, Clade GRY), beta (B.1.351, Clade GH), gamma (P.1, Clade GR), and delta (B.1.617.2, Clade G). VOCs are likely to have some effect on transmissibility, antibody evasion, and changes in therapeutic or vaccine effectiveness. However, the physiological and virological understanding of these variants remains poor. We demonstrated that these four VOCs exhibited differences in plaque size, thermal stability at physiological temperature, and replication rates. The mean plaque size of beta was the largest, followed by those of gamma, delta, and alpha. Thermal stability, evaluated by measuring infectivity and half-life after prolonged incubation at physiological temperature, was correlated with plaque size in all variants except alpha. However, despite its relatively high thermal stability, alpha’s small plaque size resulted in lower replication rates and fewer progeny viruses. Our findings may inform further virological studies of SARS-CoV-2 variant characteristics, VOCs, and variants of interest. These studies are important for the effective management of the COVID-19 pandemic.

## 1. Introduction

Several novel variants of SARS-CoV-2 have been identified since the COVID-19 pandemic began in December 2019. Although the expected mutation rate for SARS-CoV-2 is estimated to be approximately 2.4 × 10^−3^ per site per year [1], significantly more mutations and deletions across thousands of variants, all of which may alter their pathogenic potential, have already been reported [2]. The World Health Organization identifies novel SARS-CoV-2 variants of concern (VOCs) based on their potential impact on public health, according to evidence of enhanced transmissibility, increased severity, reduced antibody neutralization arising from former infection or vaccination, detection evasion ability, or decreased treatment or vaccine efficacy [3].

There are currently only four circulating SARS-CoV-2 VOCs: alpha (B.1.1.7, clade GRY), beta (B.1.351, clade GH), gamma (P.1, clade GR), and delta (B.1.617.2, clade G). They share D614G mutation, conferring increased infectivity, likely due to changes affecting the receptor binding and fusion [4,5,6,7]. N501Y mutation is also shared by alpha, beta, and gamma, increasing their receptor-binding affinity and subsequent cellular entry [8]. However, the combination of mutations could result in greater conformational changes and distinctive modifications [9]. For example, VOCs exhibit differential receptor-binding affinity. Alpha requires the most force to be detached from the receptor, followed by beta/gamma and delta [10].

## 2. Materials and Methods

### 2.1. Cell Lines and Viruses

Vero E6 cells (ATCC, CCL-1586) were maintained in Eagle’s minimal essential medium (EMEM; WELGENE) supplemented with 10% fetal bovine serum (FBS; Gibco, Carlsbad, CA, USA) and 100 U/mL penicillin-streptomycin (P/S; Gibco). Three micrograms of pCMV3-TMPRSS2-HA plasmid DNA (Cat# 40143-R001, Sino Biological, Beijing, China) was transfected into Vero E6 cells in a 6-well plate using Lipofectamine LTX and PLUS reagent (Cat# 15338-100, Invitrogen, Carlsbad, CA, USA) as per the protocol. After maintaining the cells in non-selective medium for 2 days, the cells were placed in selective medium containing 500 μg/mL of Hygromycin B (Cat# ant-hg-1, Invivogen) until un-transfected control cells completely died. The survived cells were further maintained in the medium containing 200 μg/mL of Hygromycin B and used for the plaque assay as Vero E6-TMPRSS2 cells. SARS-CoV-2 variants (NCCP number: 43,381 (alpha), 43,382 (beta), 43,388 (gamma), and 43,390 (delta)) were obtained from the National Culture Collection of Pathogens of South Korea (NCCP). The S protein mutations of all four VOCs used in this study were confirmed by RNA-seq (Alpha: N501Y, A570D, D614G, P681H, T716I, S982A, D1118H; Beta: L18F, D80A, D215G, K417N, E484K, N501Y, D614G, A701V; Gamma: E484K, D614G, V1176F; Delta: T19R, G142D, E156G, Δ157-158, L452R, T478K, D614G, P681R). All experiments involving infectious SARS-CoV-2 variants were performed in a biosafety level 3 (BSL-3) containment laboratory at the Korean Research Institute for Chemical Technology (KRICT, Daejeon, South Korea).

### 2.2. Viral Titer Determination

Plaque- and focus-forming assays were performed as previously reported with some modifications [11]. In brief, for the plaque assay, the virus was serially diluted in EMEM supplemented with 2% FBS. The cell culture medium was removed from Vero E6 or Vero E6-TMPRSS2 cells (1 × 10^5^ per 24-well) 1 day prior to the assay, before the inoculum was transferred onto triplicate cell monolayers. These cells were incubated at 37 °C for 1 h, and the inoculum was discarded before the infected cells were overlaid with 1.8% carboxymethyl cellulose in MEM. Samples were incubated at 37 °C for 3 days (Vero E6-TMPRSS2) or 4 days (Vero E6) before they were fixed and stained using 0.05% crystal violet in 1% formaldehyde. Plaque counts and size evaluations were completed using an ImmunoSpot analyzer (C.T.L).

For the focus-forming assay, Vero E6 cells (2 × 10^4^ per 96-well) were infected with diluted inoculum for 1 h and then placed in fresh medium for an additional 8 h, after which they were washed and fixed. Cells were then stained with the N-specific antibody (Cat# 40143-R001, Sino Biological) and secondary horseradish peroxidase-conjugated goat anti-rabbit IgG (Cat# 170-6515, Bio-Rad, Hercules, CA, USA). The signal was then developed using an insoluble TMB substrate (Promega, Madison, WI, USA), and the number of infected cells was counted using an ImmunoSpot analyzer. Intra- and extracellular viral RNA was quantified using quantitative RT-PCR (QuantStudio 3, Applied Biosystems, Foster City, CA, USA), which was completed using the one-step Prime script III qRT-PCR mix (Takara). Viral RNA was detected using a 2019-nCoV-N1 probe (Cat#10006770, Integrated DNA Technologies, Coralville, IA, USA).

### 2.3. Statistical Analysis

Mean values for the plaque size and half-life assays were compared using one-way ANOVA in GraphPad Prism 8.0 software (GraphPad Software, San Diego, CA, USA). Statistical significance was set at *p* < 0.05.

## 3. Results

Here, we noted that each of these variants presented with different plaque sizes in Vero E6 cells (Figure 1A). The mean plaque size of beta was largest, followed by those of gamma, delta, and alpha (Figure 1B). A similar result of different plaque sizes was obtained in Vero E6-TMPRSS2 cells (Figure 1C,D). While there are numerous determinants of plaque size, we hypothesized that changes in the receptor-binding affinity, thermal stability, and replication rate of these viruses were likely factors. Based on the previously reported receptor-binding affinity data described above, we examined the thermal stability and replication rate of each of these VOCs. If a variant experiences increased stability at physiological temperatures and produces more progeny viruses, more infectious viral particles are yielded, infecting more cells and increasing the plaque size. Given this, we first assessed the thermal stability of these variants in culture media incubated at different temperatures over an 8-h period. Their infectivity was then measured using a focus-forming assay. Owing to the short incubation time following viral infection, this assay allowed for more precise assessments of virion infectivity, as it is less affected by other factors associated with viral replication.

Of the four VOCs, the beta variant was the most stable at 4, 24, and 37 °C (Figure 2A). We then evaluated the relative stability by measuring changes in their infectivity following prolonged incubation (2, 4, 8, 12, and 24 h) in solutions at physiological temperatures using the focus-forming assay. As expected, beta exhibited the highest thermal stability (Figure 2B), with a half-life approximately twice that of gamma or delta (Figure 2C). These results indicated a correlation between thermal stability and plaque size in all variants except alpha. Interestingly, despite the small plaque size of alpha, its half-life was relatively long, suggesting an alternative mechanism.

Next, we examined the viral replication rates of these VOCs. Vero E6 cells were infected with VOCs at the same multiplicity of infection (MOI: 0.1). Plaque forming assay and qRT-PCR were then used to assess infectious viral particle numbers and viral RNA concentrations, respectively. Alpha had fewer infectious viral particles than the other variants, although there were no significant differences in viral RNA copy number (Figure 3A,B). Moreover, the intracellular viral RNA concentrations of alpha and gamma were significantly lower than those of the other VOCs (Figure 3C). These results indicate that the viral replication rate of alpha is likely low, contributing to its small plaque size.

## 4. Discussion

It is important to analyze these VOCs in terms of both their clinical pathology and virology, as this will help us better understand their increased virulence. For example, plaque size may be associated with contagiousness or viral transmission. Here, we noted that most of our VOCs presented with changes in plaque size. In addition, these changes were largely mirrored by changes in virion stability at physiological temperatures and the concentration of infectious viral particles. Correlation analyses suggested that there is a strong link between plaque size and thermal stability in these VOCs and that the relatively large plaque size of beta may account for its increased thermal resistance. This increased stability may contribute to its pathobiology and transmission, requiring further studies of the viral titer and case fatality rate in humans. Notably, a recent study reported that severity, criticality, and fatality of beta were higher than alpha in COVID-19 patients [12]. Conversely, although alpha presented with increased thermal stability, this did not translate to increased plaque size. This reduced plaque size may be explained by the reduced number of infectious particles produced by alpha under these conditions. Further studies will be required to identify additional determinants of plaque size, including functional mutations, interactions with host factors, and environmental composition. In addition, variants of interest and other variants should also be investigated to effectively control their spread.

## Figures and Tables

**Figure 1 viruses-14-00055-f001:**
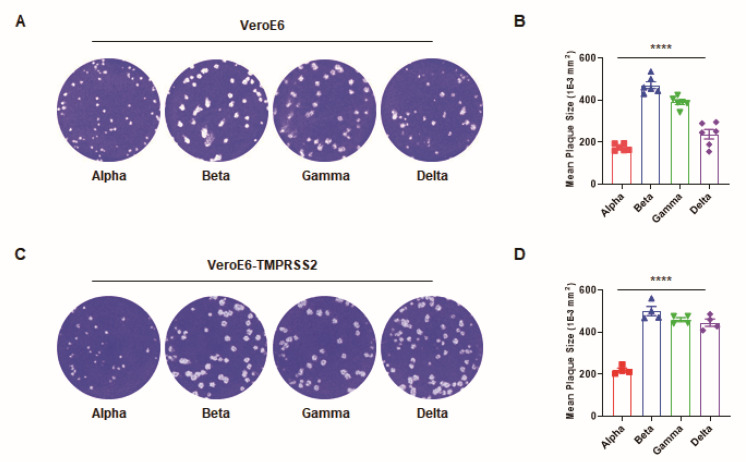
Comparison of the plaque size of four SARS-CoV-2 variants. (**A**) Representative images of the plaque forming assay for each variant of concern in Vero E6 cells. (**B**) The mean plaque size (1 × 10^−3^ mm^2^) of each variant. (**C**) The images of the plaque forming assay for each variant of concern in Vero E6-TMPRSS2 cells. (**D**) The mean plaque size of each variant. All error bars indicate the standard errors of the mean. Mean values for the plaque size were compared using one-way ANOVA in GraphPad Prism 8.0 software. Statistical significance was set at *p* < 0.05. (****, *p* < 0.0001).

**Figure 2 viruses-14-00055-f002:**
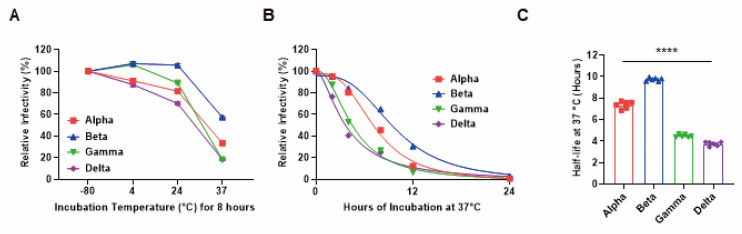
Different thermal stability of four SARS-CoV-2 variants. (**A**) The relative infectivity of each variant after incubation at 4, 24, or 37 °C for 8 h as evaluated via the focus-forming assay. (**B**) Nonlinear regression of the relative infectivity of each variant following prolonged incubation (2, 4, 8, 12, and 24 h) at physiological temperatures (37 °C). (**C**) Average half-life values for each variant at 37 °C. All error bars indicate the standard errors of the mean. Mean values for the half-life were compared using one-way ANOVA in GraphPad Prism 8.0 software. Statistical significance was set at *p* < 0.05 (****, *p* < 0.0001).

**Figure 3 viruses-14-00055-f003:**
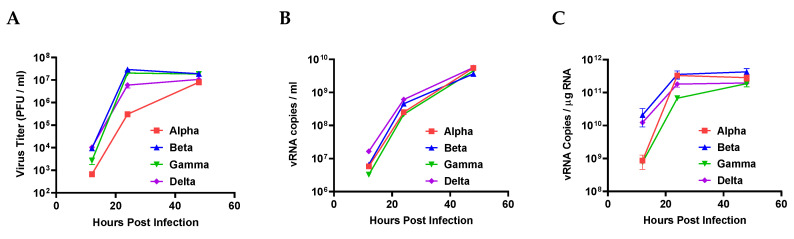
Viral replication rate of four SARS-CoV-2 variants. (**A**) The infectivity of the progeny viruses was evaluated via plaque forming assay completed at 12, 24, and 48 h post infection. (**B**) Extracellular and (**C**) intracellular viral RNA were assessed by qRT-PCR with SARS-CoV-2 NP probes. All error bars indicate the standard errors of the mean.

## Data Availability

The datasets used and/or analyzed during the current study are available from the corresponding author on reasonable request.

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
