# Peer review of "Comparison of Plaque Size, Thermal Stability, and Replication Rate among SARS-CoV-2 Variants of Concern"

_viruses, 2021, doi:10.3390/v14010055_

Round 1

Reviewer 1 Report

The article Comparison of Plaque Size, Thermal Stability, and Replication Rate among SARS-CoV-2 Variants of Concern written by Gi Uk Jeong et al. presented intresting data.  The comments are provided in manuscript.  Thanking you  

Author Response

The article Comparison of Plaque Size, Thermal Stability, and Replication Rate among SARS-CoV-2 Variants of Concern written by Gi Uk Jeong et al. presented intresting data.  The comments are provided in manuscript.  Thanking you  

Response: We thank the reviewer for his/her consideration and helpful comments in improving our manuscript. We have responded to each of the critiques and incorporated changes in the appropriate sections of the revised manuscript. A point-by-point discussion of the comments from the reviewer is provided below.

  1. How this will help in treatment

Response: We think that our findings are helpful for understanding molecular mechanisms of the virulence and transmissibility of all four variants to guide therapeutic strategies. For example, a recent study reported that severity, criticality, and fatality of beta were higher than alpha in COVID-19 patients [12]. This might be correlated with our data that present the largest plaque size of beta with higher thermal stability and replication rate than those of alpha. We have revised the manuscript (Lines 128-129) and included additional reference.

  1. Please describe the mutation present in those VOCs used for the analysis. All four VOCs are mutating very fast and we need to understand if these findings are specific to strain or generic for VOCs.

Response: Thank you for pointing this out. We have now included the mutations in S protein of all four VOCs used in this study by RNA-seq. Lines 64-67 in the revised manuscript.

  1. Please explain more why the size are different for different variants??

What could be possible reason for this?? as per current data Delta is having highest translatability, does it have any role in replication ??

Response: Thank you for your suggestion. There can be other contributions to different plaque sizes of VOCs such as the RBD-binding affinity [10] and the membrane fusion (Reference DOI: 10.1126/science.abl9463). However, in this study, we found that plaque size changes were largely mirrored by changes in virion stability at the physiological temperature and the concentration of infectious viral particles. Unfortunately, we were not able to find any published data about the highest translatability of delta.

  1. Please explain how this data would be useful for further understanding about this virus or playing any role in standard assays performed. Using example of other virus's can you correlate the stability and plaque size.

Response: As described above, we think that our data are useful for better understanding molecular mechanisms of the virulence and transmissibility of all four variants. For example, a recent study reported that severity, criticality, and fatality of beta were higher than alpha in COVID-19 patients [12]. This might be correlated with our data that present the largest plaque size of beta with higher thermal stability and replication rate than those of alpha. We have revised the manuscript (Lines 128-129) and included additional reference.

Reviewer 2 Report

The authors compare plaque size thermal stability and replication rate among  alpha, beta, gamma and delta SARS-CoV2 VOCs. They find that the beta variant had the largest mean plaque size, and that thermal stability was correlated with plaque size in all variants except alpha.

The work is interesting and deserves to be published after some revisions. The discussion should be more attractive by correlating the in vitro results obtained in this work with the known epidemiology observed in patients. The authors should discuss and correlate their results with the available data previously published in patients according to the VOCs.

It would be interesting to compare plaque size with viral loads found in patients / the possible differences in the R0 (reproduction rate)/ the inter-human transmissibility according to the VOC: are they correlated?

It also would be interesting to compare the results of thermal stability found in this work with the seasonal peaks of the incidence of COVID19 according to the VOCs, by taking into account the differences in temperature according to the countries.

Author Response

Comments and Suggestions for Authors

The authors compare plaque size thermal stability and replication rate among alpha, beta, gamma and delta SARS-CoV2 VOCs. They find that the beta variant had the largest mean plaque size, and that thermal stability was correlated with plaque size in all variants except alpha.

Response: We thank the reviewer for his/her consideration and helpful comments in improving our manuscript. We have responded to each of the critiques and incorporated changes in the appropriate sections of the revised manuscript. A point-by-point discussion of the comments from the reviewer is provided below.

The work is interesting and deserves to be published after some revisions. The discussion should be more attractive by correlating the in vitro results obtained in this work with the known epidemiology observed in patients. The authors should discuss and correlate their results with the available data previously published in patients according to the VOCs.

Response: Thank you for your comment. A recent study reported that severity, criticality, and fatality of beta were higher than alpha in COVID-19 patients [12]. This might be correlated with our data that present the largest plaque size of beta with higher thermal stability and replication rate than those of alpha. We have revised the discussion part (Lines 128-129) and included additional reference.

It would be interesting to compare plaque size with viral loads found in patients / the possible differences in the R0 (reproduction rate)/ the inter-human transmissibility according to the VOC: are they correlated?

Response: We appreciate your suggestion. A previous study reported that the delta variant had a higher viral load than alpha and beta in nasopharyngeal samples from COVID-19 patients (Reference DOI: 10.1016/j.jinf.2021.08.027). But, the viral load growth rate of delta was lower than alpha in COVID-19 individuals (Reference DOI: 10.1016/S1473-3099(21)00648-4). Thus, we think that it might be hard to appropriately compare viral loads of the VOCs in time with the plaque size.

It also would be interesting to compare the results of thermal stability found in this work with the seasonal peaks of the incidence of COVID19 according to the VOCs, by taking into account the differences in temperature according to the countries.

Response: Thank you for this point. We tried to find some correlation between thermal stability and each temperatures of the incidence of VOCs in UK, South Africa, Brazil, and India. Unfortunately, we were not able to find any clues.
